# Analysing the sensitivity of pollen based land-cover maps to different auxiliary variables

Behnaz Pirzamanbein<sup>1,2</sup>, Anneli Poska<sup>3,4</sup>, and Johan Lindström<sup>1</sup>

<sup>1</sup>Centre for Mathematical Sciences, Lund University, Sweden

<sup>2</sup>Centre for Environmental and Climate Research, Lund University, Sweden

<sup>3</sup>Department of Physical Geography and Ecosystems Analysis, Lund University, Sweden

<sup>4</sup>Institute of Geology, Tallinn University of Technology, Estonia

Correspondence to: Behnaz Pirzamanbein (behnaz@maths.lth.se)

Abstract. Realistic depictions of past land cover are needed to investigate prehistoric environmental changes and anthropogenic impacts. However, observation based reconstructions of past land cover are rare. Recently Pirzamanbein et al. (2015, arXiv:1511.06417) developed a statistical interpolation method that produces spatially complete reconstructions of past land cover from pollen assemblage. These reconstructions incorporate a number of auxiliary datasets raising questions regarding both the method's sensitivity to the choice of auxiliary data and the unaffected transmission of observational data.

Here the sensitivity of the method is examined by performing spatial reconstructions for northern Europe during three time periods (1900 CE, 1725 CE and 4000 BCE), based on irregularly distributed pollen based land cover, available for ca 25% of the area, and different auxiliary datasets. The spatially explicit auxiliary datasets considered include the most commonly utilized sources of the past land-cover data — estimates produced by a dynamic vegetation (DVM) and anthropogenic land-

cover change (ALCC) models — and modern elevation. Five different auxiliary datasets were considered, including different 10 climate data driving the DVM and different ALCC models. The resulting reconstructions were evaluated using deviance information criteria and cross validation for all the time periods. For the recent time period, 1900 CE, the different land-cover reconstructions were also compared against a present day forest map.

The tests confirm that the developed statistical model provides a robust spatial interpolation tool with low sensitivity to 15 differences in auxiliary data and high capacity to un-distortedly transmit the information provided by sparse pollen based observations. Further, usage of auxiliary data with high spatial detail improves the model performance for the areas with complex topography or where observational data is missing.

# 1 Introduction

5

The importance of terrestrial land cover for the global carbon cycle and its impact on the climate system is well recognized (e.g. Claussen et al., 2001; Brovkin et al., 2006; Arneth et al., 2010; Christidis et al., 2013). Many studies have found large 20 climatic effects associated with changes in land cover. Forecast simulations evaluating the effects of human induced global warming predict a considerable amplification of future climate change, especially for Arctic areas. The amplification is due to a number of biogeophysical and chemical feedbacks brought by the northward advancement of boreal shrub and treeline (Zhang

5

et al., 2013; Richter-Menge et al., 2011; Chapman and Walsh, 2007; Koenigk et al., 2013; Miller and Smith, 2012). The past anthropogenic deforestation of the temperate zone in Europe was lately demonstrated to have an impact on regional climate similar in amplitude to present day climate change (Strandberg et al., 2014). However, studies on the effects of vegetation and land-use changes on past climate and carbon cycle often report considerable differences and uncertainties in their model predictions (de Noblet-Ducoudré et al., 2012; Olofsson, 2013).

One of the reasons for such widely diverging results could be the differences in past land-cover descriptions used by climate modellers. Possible land-cover descriptions range from static present-day land cover (Strandberg et al., 2011), over simulated potential natural land cover from dynamic (or static) vegetation models (DVMs) (e.g. Brovkin et al., 2002; Hickler et al., 2012), to past land-cover scenarios combining DVM derived potential vegetation with estimates of anthropogenic land-cover change

- 10 (ALCC) (Strandberg et al., 2014; Pongratz et al., 2008; de Noblet-Ducoudré et al., 2012). Differences in input climates, inherent mechanistic and parametrisation differences of DVMs (Prentice et al., 2007; Scheiter et al., 2013), and significant variation in land-use estimates between the existing ALCC scenarios (e.g. Kaplan et al., 2009; Pongratz et al., 2008; Klein Goldewijk et al., 2011; Gaillard et al., 2010) further contribute to the differences in past land-cover descriptions. These differences can lead to largely diverging estimates of past land-cover dynamics even when the most advanced models are used (Strandberg and 2014).
- 15 et al., 2014; Pitman et al., 2009).

The palaeoecological observation based land-cover reconstructions (LCR) recently published by Pirzamanbein et al. (2014, 2015) were designed to overcome the above described problems. And to provide an alternative, observation based, land-cover description applicable for a range of studies on past vegetation and its interactions with climate, soil and humans. These reconstructions use the pollen based land-cover composition (PbLCC) published by Trondman et al. (2015) as a source of

- 20 information on past land-cover composition. The PbLCC are point estimates, depicting the land-cover composition of the area surrounding each of the studied sites. To fill the gaps between these observations and to acquire a spatially continuous landcover reconstruction, spatial interpolation is necessary. Conventional interpolation methods might struggle when handling noisy, spatially heterogeneous data (Heuvelink et al., 1999; De Knegt et al., 2010), but alternative statistical methods for handling spatially structured data exist (e.g. Gelfand et al., 2010; Blangiardo and Cameletti, 2015).
- In Pirzamanbein et al. (2015) a statistical model based on Gaussian Markov Random Fields (GMRFs, Lindgren et al., 2011; Rue and Held, 2004) was developed to provide a reliable, computationally effective and freeware based spatial interpolation technique. The resulting statistical model combines PbLCC data with auxiliary datasets; e.g. DVM output, ALCC scenarios, and elevation; to produce reconstructions of past land cover. The auxiliary data is subject to the differences and uncertainties outlined above and the choice of auxiliary data could influence accuracy of the statistical model. This study aims at determining
- 30 the robustness of the model; to evaluate its capacity to un-distortedly recover information provided by PbLCC observations on past vegetation composition; and to analyse the models sensitivity to different auxiliary datasets.

15

# 2 Material and methods

The studied area covers temperate, boreal and alpine-arctic biomes of central and northern Europe ( $45^{\circ}$ N to  $71^{\circ}$ N and  $10^{\circ}$ W to  $30^{\circ}$ E). The Pollen based land-cover composition (PbLCC) published in Trondman et al. (2015) consists of proportions of coniferous forest (CF), broadleaved forest (BF) and un-forested land (UF) presented as gridded ( $1^{\circ} \times 1^{\circ}$ ) data points placed irregularly across northern-central Europe. Altogether 175 grid cells containing observational data were available for 1900 CE, 181 for 1725 CE, and 196 for the 4000 BCE time-period (Figure 1, column 2).

5 Four different model derived datasets depicting past land cover were considered as potential auxiliary datasets. In each case potential natural vegetation (PNV) composition estimated by the dynamic vegetation model (DVM) LPJ-GUESS (Lund-Potsdam-Jena General Ecosystem Simulator; Smith et al., 2001; Sitch et al., 2003) is combined with an ALCC scenario to adjust for human land use (see Pirzamanbein et al., 2014, for more detail):

K-L<sub>RCA3</sub>: Combines the ALCC scenario KK10 (Kaplan et al., 2009) and the PNV composition from LPJ-GUESS. Climate

- forcing for the DVM was derived from RCA3 (Rossby Centre Regional Climate Model, Samuelsson et al., 2011) at annual time and  $0.44^{\circ} \times 0.44^{\circ}$  spatial resolution (Figure 1, column 3),
  - **K-L<sub>ESM</sub>:** Combines the ALCC scenario KK10 and the PNV composition from LPJ-GUESS. For this dataset, the climate forcing for the DVM was derived from the Earth System Model (ESM; Mikolajewicz et al., 2007) at centennial time and  $5.6^{\circ} \times 5.6^{\circ}$  spatial resolution. To interpolate data into annual time and  $0.5^{\circ} \times 0.5^{\circ}$  spatial resolution climate data from 1901–1930 CE provided by the Climate Research Unit (CRU) was used (Figure 1, column 4),
  - **H-L<sub>RCA3</sub>:** Combines the ALCC scenario from the History Database of the Global Environment (HYDE; Klein Goldewijk et al., 2011) and the PVN composition from LPJ-GUESS with RCA3 climate forcing (Figure 1, column 5),
  - H-L<sub>ESM</sub>: Combines the ALCC scenario from HYDE and the PVN composition from LPJ-GUESS with ESM climate forcing (Figure 1, column 6).
- In addition, elevation data used in modelling was obtained from the Shuttle Radar Topography Mission (SRTM<sub>elev</sub>, Becker et al., 2009) (Figure 1, column 1 row 2).

Finally, a modern forest map based on data from the European Forest Institute (EFI) is used for evaluation of the model's performance for the 1900 CE time period. The EFI forest map (EFI-FM) is based on a combination of satellite data (NOAA-AVHRR) and national forest-inventory statistics from 1990—2005 (Päivinen et al., 2001; Schuck et al., 2002) (Figure 1, column 1 row 1).

All above described sets of auxiliary data were up-scaled to  $1^{\circ} \times 1^{\circ}$  spatial resolution, matching the pollen based reconstructions, before usage as model input.