# Peer review of "Analysing the sensitivity of pollen based land-cover maps to different auxiliary variables"

_Climate of the Past, 2017_

## Referee Comment (RC1) · Anonymous Referee #1 · 27 Jun 2017

Pirzamanbein et al. present a statistical method for producing past land cover reconstructions from pollen and model data. More specifically, they explore present and test a statistical model that can be used for linking pollen-based land cover data with different auxiliary variables, such as the past vegetation simulated with dynamic vegetation modeling and anthropogenic land cover change modeling. The idea of combining pollen-based land cover data with auxiliary data derived from various simulation approaches is interesting. The paper has a clear focus and it is for most parts clearly presented and well illustrated, deserving a publication.

My main comment is, however, that the authors should reconsider whether Climate of the Past is the right and best forum for this paper. It is certainly true that past land cover patterns are important for understanding past climates, but this paper is really not

presented from the palaeoclimatological perspective. Its emphasis is in the use of the statistical model for land-cover compositions, and the mathematical basis of this model is presented and tested in detail on pages 4-8. As a consequence, the paper is mostly a methodological description of the model, including its testing. There is barely any description or discussion why this model might be important in palaeoclimatology and I fear that the number of palaeoclimatologists interested in the details of this particular statistical model is very limited. My view is that there are other journals, which would be more suitable for this paper, for example Environmetrics, Biometrics, Computers and Geosciences or possibly Journal of Applied Ecology.

More detailed remarks.

-page 1 line 2: "However, observation based reconstructions of past land cover are rare". This paper deals mostly with land covers before human-made observations, so a better term instead of "observations" would be "proxy-based reconstructions" - it is stated in the abstract that five different auxiliary datasets were considered in the study. However, on page 3 the authors write that "Four different model derived datasets. . .were considered as potential auxiliary datasets". This seems contradictory. I believe that the fifth auxiliary dataset is the elevation dataset? In any case, it would be best to amend the wording either in the abstract or in the method description. -I agree with the authors when they write that "The final land-cover reconstructions achieved by fitting the models to the observed PbLCC are very similar. . .". This is demonstrated by maps in Figs. 4-6. But I find this outcome surprising because earlier in the paper (page 8 lines 15-20 and Fig. 1) it is stated that the available auxiliary datasets exhibit large variation in the extent of coniferous and broadleaved forests, and un-forested areas for all of the studied time periods. It is hard to understand how it is possible that when these different auxiliary datasets, showing such large variation, are combined with one and the same pollen—based dataset, the resulting land-cover reconstructions are nearly identical. This can lead to a skeptical view about the performance of the statistical model presented in the paper. -page 15 line 11: "The performance of the statistical

model. . .to reconstruct the pollen based observations was tested. . ." The model is not used "to reconstruct the pollen based observations". It would be more correct to write that the statistical model was used to test the sensitivity of the pollen-based land cover reconstructions to the use of different auxiliary datasets". -the possible palaeoclimatic importance of the results presented is totally lacking from the Discussion from the Conclusions. -figures and captions should be made more user-friendly. The caption of Fig. 1 is very tedious to read with many abbreviations and Fig. 2 is hard to understand given the short and uninformative caption. Figure 9 is only briefly mentioned in the text and the caption is uninformative.

---

## Referee Comment (RC2) · Anonymous Referee #2 · 3 Jul 2017

This is a well-constructed paper which clearly compares different methods of generating past land cover maps from partial data derived from pollen records, and merits publication somewhere. The paper uses auxiliary data from other landcover reconstructions (e.g. Dynamic Vegetation models or population-based land cover models) to inform extrapolation, which apparently improves performance but also introduces new assumptions, which are not clearly addressed. To this non-expert reader, an element of circularity seemed to be present in some of the data combinations – this is quite possibly my misunderstanding, but given the journal's audience could usefully be refined.

My principal comment is that, like the first referee, I'm not sure that Climate of the Past is the right place for this article. The content focuses on model choices and assumptions, and although the relevance to palaeoclimate is clearly stated, it's not well brought out and there is no clear take-home message of interest or use to a palaeoclimate scientist. As is, I don't see this paper being of much relevance/appeal to most readers of the journal, and therefore it might get lost to some extent.

The paper lacks a clear conclusion relevant to the wider community – of the different combinations tested, which is recommended for use by future researchers? What is the best strategy for multiple time periods, and is the recommendation likely to extrapolate beyond Europe or is this something that needs to be carried out in each area and for each time period? How significant are the improvements in model output from adding the auxiliary data? I can see the numbers in tables 3 and 4, but I find it hard to judge what they mean in terms of actual improvement gained, and whether that is actually worthwhile given that including the auxiliary data generally also involves adding more assumptions to the reconstruction, thereby increasing other kinds of uncertainty.

A couple of minor points: 1) I was not convinced by the testing method of comparing vegetation reconstructed for 1900CE with modern EFI data, since a great deal has happened to land cover and forestry in Europe in the last 100 or so years, yet the authors treat the comparison as if it is like for like. That may be a valid assumption, but I'd expect to see that considered overtly rather than assumed in a paper like this. 2) A table of the algebraic symbols used would be useful – at the moment, terms are not always defined at time of first introduction, or easy to retain, especially as many single symbols refer to matrices rather than individual values.

---

## Editor Comment (EC1) · N. Combourieu Nebout (Editor) · 10 Jul 2017

Dear authors,

We have now received two reviews of your paper. All two reviewers ask you many questions that need your response. You have now to post your replies to all the comments on the discussion forum. After your response I will send my decision on your paper.

With my best regards.

Nathalie Combourieu-Nebout

---

## Author Comment (AC1) · 7 Aug 2017

The authors would like to thank the reviewer for making the effort to read our manuscript and provide good constructive feedback, and for appreciation of our work.

The responses was uploaded in the form of a supplement.

Please also note the supplement to this comment:
https://www.clim-past-discuss.net/cp-2017-51/cp-2017-51-AC1-supplement.pdf

---

## Author Comment (AC2) · 7 Aug 2017

**General:**

The aim of this paper is to present and evaluate a methodology that produces spatially explicit land cover reconstructions from pollen based proxy data. The methods sensitivity to different auxiliary variables is tested, and shown to be very low. Finally, we provide past land-cover maps that can be used directly in the climate models.

Although the paper is somewhat mathematical we feel it to be relevant for climate of the past since: 1) Palaeoecological proxies, such as pollen, are valuable source of information on past environmental conditions, but hardly applicable by climate modellers as input in their original format, and therefore heavily underused; 2) We present a general way of extracting spatially continuous land cover from pollen proxy data producing spatially explicit proxy based land cover maps directly usable in climate models; and 3) The resulting reconstructions of past land-cover for Europe during two important time windows are provided as auxiliary material in the paper. These pollen based land cover reconstructions could be used in climate models to facilitate mechanistic studies on past climate-land cover relationships.

To clarify these points, text (outlining the points above) have been added to the abstract, introduction, results and discussion, and conclusions sections.

**Reviewer 2:**

This is a well-constructed paper which clearly compares different methods of generating past land cover maps from partial data derived from pollen records, and merits publication somewhere. The paper uses auxiliary data from other land-cover reconstructions (e.g. Dynamic Vegetation models or population-based land cover models) to inform extrapolation, which apparently improves performance but also introduces new assumptions, which are not clearly addressed. To this non-expert reader, an element of circularity seemed to be present in some of the data combinations  this is quite possibly my misunderstanding, but given the journal's audience could usefully be refined.

My principal comment is that, like the first referee, I'm not sure that Climate of the Past is the right place for this article. The content focuses on model choices and assumptions, and although the relevance to palaeoclimate is clearly stated, it's not well brought out and there is no clear take-home message of interest or use to a palaeoclimate scientist. As is, I don't see this paper being of much relevance/appeal to most readers of the journal, and therefore it might get lost to some extent.

**Reply:** The abstract, as well as initial paragraphs of the sections on "results and discussion" and "conclusions" have been updated to illustrate how the method and results (i.e. publicly available datasets of land-cover reconstructions) can be used to facilitate the mechanistic studies on past climate-land cover relationships. These changes are discussed in more detail in the general comments.

- The paper lacks a clear conclusion relevant to the wider community of the different combinations tested, which is recommended for use by future researchers?

  **Reply:** An important point of this evaluation study is the robustness of the method to different auxiliary datasets. To clarify this important feature the following text has been added to Page 15, line 16:

  "... remains unchanged. *Therefore, the model can provide reliable results using a variety of land cover data sets that capture important spatial patterns from vegetation models and past human land use, absent good covariates elevation can be used as the only auxiliary dataset. An important feature of the suggested model is the estimation of different weights for each of the auxiliary datasets (see table 2), thus capturing the spatial patterns and not the absolute values in the auxiliary datasets. Our validations indicate that auxiliary datasets obtained using different climatic drivers produce very similar reconstructions, which are all close to the pollen based proxy data.*"

- What is the best strategy for multiple time periods, and is the recommendation likely to extrapolate beyond Europe or is this something that needs to be carried out in each area and for each time period?

  **Reply:** We have high hopes that the method should be generally applicable across a broad range of regions and time periods. The current status of pollen proxy data has been expanded on by additional text on Page 15, line 19:

  "... (e.g. Gaillard et al., 2010; Strandberg et al., 2015). The results also indicate that the model has a very good performance and will be very useful for large-scale, continental reconstructions of past land cover. The spatial model tested in this paper can provide an important tool to generate regional to global scale land-cover maps based on proxy data. Such, pollen based past land cover reconstructions with global coverage are currently produced by the PAGES (Past Global changES) LandCover6k initiative [1] for most of globe."

- How significant are the improvements in model output from adding the auxiliary data? I can see the numbers in tables 3 and 4, but I find it hard to judge what they mean in terms of actual improvement gained, and whether that is actually worthwhile given that including the auxiliary data generally also involves adding more assumptions to the reconstruction, thereby increasing other kinds of uncertainty.

  **Reply:** This point is partially related to the question regarding the model's robustness to different auxiliary datasets raised above and by reviewer 1. Using more spatially explicit auxiliary datasets is likely to
* * *
[1] www.pastglobalchanges.org/ini/wg/landcover6k/intro

help with fine scale detail (e.g. effects of coastal and mountainous climates). In addition to changes at Page 15, line 16 outlined above we have also added a paragraph to page 10, line 12 and a new table (Table 3):

"... datasets used. *At first the similarity among the reconstructions might seem contradictory, but recall that the model allows for, and estimates, different weighting (the regression coefficients, $\boldsymbol{\beta}$:s) for each of the auxiliary datasets. Thus, the resulting reconstruction do not rely on the absolute values in the auxiliary datasets, only their spatial patterns; Table 3 illustrates the substantial discrepancies in the estimated coefficients, $\boldsymbol{\beta}$. Although ...*"

A couple of minor points:

1. I was not convinced by the testing method of comparing vegetation reconstructed for 1900 CE with modern EFI data, since a great deal has happened to land cover and forestry in Europe in the last 100 or so years, yet the authors treat the comparison as if it is like for like. That may be a valid assumption, but I'd expect to see that considered overtly rather than assumed in a paper like this.

   **Reply:** To clarify this issue we have added extra text to Page 10 , line 10

   "*Although a temporal misalignment exists between the PbLCC data for the 1900 CE time period (based on pollen data from 1850 to the present) and the EFI-FM (inventory and satellite data from 1990-2005); EFI-FM provides the best complete and consistent land cover map of Europe for present time, making it a reasonable choice for the comparison. The main differences between the EFI-FM and the PbLCC data for the 1900 CE time period are: 1) lower abundance of broadleaved forests for most of Europe, 2) higher abundance of coniferous forest in Sweden and Finland, and 3) higher abundance of unforested land in North Norway in the EFI-FM data than in the PbLCC data (Pirzamanbein et al. 2015).*"

2. A table of the algebraic symbols used would be useful at the moment, terms are not always defined at time of first introduction, or easy to retain, especially as many single symbols refer to matrices rather than individual values.

   **Reply:** A list of notations has been added as a new table (Table 1) for clarification.

---

## Author Comment (AC3) · 7 Aug 2017

**Analysing the sensitivity of pollen based land-cover maps to different auxiliary variables**

Behnaz Pirzamanbein[1,2], Anneli Poska[3,4], and Johan Lindström[1]

[1]Centre for Mathematical Sciences, Lund University, Sweden
[2]Centre for Environmental and Climate Research, Lund University, Sweden
[3]Department of Physical Geography and Ecosystems Analysis, Lund University, Sweden
[4]Institute of Geology, Tallinn University of Technology, Estonia

*Correspondence to:* Behnaz Pirzamanbein (behnaz@maths.lth.se)

**Abstract.** Realistic depictions of past land cover are needed to investigate prehistoric environmental changes and  to determine the scale of anthropogenic deforestation on environment and to study long term land cover-climate feedbacks. However, observation based reconstructions of past land cover are rare while commonly used modelled based reconstructions exhibit considerable differences and give diverging results when used by climate modellers to study the effects of past land-cover dynamics on climate. Recently Pirzamanbein et al. (2015, arXiv:1511.06417) developed a statistical interpolation method that produces spatially complete reconstructions of past land cover from pollen assemblage. These reconstructions incorporate a number of auxiliary datasets raising questions regarding both the method's sensitivity to the choice of auxiliary data and the unaffected transmission of observational data.

[revised manuscript text omitted]

30  technique. The resulting statistical model combines PbLCC data with auxiliary datasets; e.g. DVM output, ALCC scenarios, and elevation; to produce reconstructions of past land cover. The auxiliary data is subject to the differences and uncertainties outlined above and the choice of auxiliary data could influence accuracy of the statistical model.  The major objectives of this paper are: 1) to draw attention of climate modelling community to a novel set of spatially explicit proxy based land-cover reconstructions suitable for climate modelling; 2) to present and test the robustness of

35  statistical spatial interpolation model for proxy based reconstructions developed by Pirzamanbein et al. (2015); 3) to evaluate

its capacity to un-distortedly recover information provided by PbLCC  proxy data on past vegetation composition; and 4) to analyse the models sensitivity to different auxiliary datasets.

**2 Material and methods**

The studied area covers temperate, boreal and alpine-arctic biomes of central and northern Europe ($45°$N to $71°$N and $10°$W to $30°$E). The Pollen based land-cover composition (PbLCC) published in Trondman et al. (2015) consists of proportions of coniferous forest (CF), broadleaved forest (BF) and un-forested land (UF) presented as gridded ($1° \times 1°$) data points placed irregularly across northern-central Europe. Altogether 175 grid cells containing  proxy data were available for 1900 CE, 181 for 1725 CE, and 196 for the 4000 BCE time-period (Figure 1, column 2).

Four different model derived datasets, depicting past land cover, along with elevation (based on SRTM data) were considered as potential auxiliary datasets. In each case potential natural vegetation (PNV) composition estimated by the dynamic vegetation model (DVM) LPJ-GUESS (Lund-Potsdam-Jena General Ecosystem Simulator; Smith et al., 2001; Sitch et al., 2003) is combined with an ALCC scenario to adjust for human land use (see Pirzamanbein et al., 2014, for more detail):

**K-L$_{RCA3}$:** Combines the ALCC scenario KK10 (Kaplan et al., 2009) and the PNV composition from LPJ-GUESS. Climate forcing for the DVM was derived from RCA3 (Rossby Centre Regional Climate Model, Samuelsson et al., 2011) at annual time and $0.44° \times 0.44°$ spatial resolution (Figure 1, column 3),

**K-L$_{ESM}$:** Combines the ALCC scenario KK10 and the PNV composition from LPJ-GUESS. For this dataset, the climate forcing for the DVM was derived from the Earth System Model (ESM; Mikolajewicz et al., 2007) at centennial time and $5.6° \times 5.6°$ spatial resolution. To interpolate data into annual time and $0.5° \times 0.5°$ spatial resolution climate data from 1901–1930 CE provided by the Climate Research Unit (CRU) was used (Figure 1, column 4),

**H-L$_{RCA3}$:** Combines the ALCC scenario from the History Database of the Global Environment (HYDE; Klein Goldewijk et al., 2011) and the PVN composition from LPJ-GUESS with RCA3 climate forcing (Figure 1, column 5),

**H-L$_{ESM}$:** Combines the ALCC scenario from HYDE and the PVN composition from LPJ-GUESS with ESM climate forcing (Figure 1, column 6).

In addition, elevation data used in modelling was obtained from the Shuttle Radar Topography Mission (SRTM$_{elev}$, Becker et al., 2009) (Figure 1, column 1 row 2).

Finally, a modern forest map based on data from the European Forest Institute (EFI) is used for evaluation of the model's performance for the 1900 CE time period. The EFI forest map (EFI-FM) is based on a combination of satellite data (NOAA-AVHRR) and national forest-inventory statistics from 1990—2005 (Päivinen et al., 2001; Schuck et al., 2002) (Figure 1, column 1 row 1).

All above described sets of auxiliary data were up-scaled to $1° \times 1°$ spatial resolution, matching the pollen based reconstructions, before usage as model input.

[Figure]

**Figure 1.** Data used in the modelling. The first column shows (from top to bottom) the EFI forest map, SRTM$_{elev}$, and the colorkey for the land-cover compositions, coniferous forest (CF), broadleaved forest (BF) and unforested land (UF). The remaining columns gives (from left to right) the pollen based land-cover composition (PbLCC, Trondman et al., 2015) and the four model based compositions that could be used as covariates: K-L$_{RCA3}$, K-L$_{ESM}$, H-L$_{RCA3}$, and H-L$_{ESM}$. Here K/H indicates KK10 (Kaplan et al., 2009) or HYDE (Klein Goldewijk et al., 2011) land use scenarios and L$_{RCA3}$/L$_{ESM}$ indicates vegetation model driven by climate from the Rossby Centre Regional Climate Model (Samuelsson et al., 2011) or Earth System Model (Mikolajewicz et al., 2007). The three rows representing (from top to bottom) the time periods 1900 CE, 1725 CE, and 4000 BCE.

**2.1 Statistical model for land-cover compositions**

A Bayesian hierarchical model (Figure 2) is used to model the PbLCC data; notation used in the model is summarised in Table 1. For each component of PbLLC, we assume an underlying compositional vector describing the proportions of land cover: coniferous forest, broadleaved forest and un-forested land. The effect of covariates and spatial structure are incorporated in the underlying compositional vector.

[Figure]

**Figure 2.** Hierarchical graph describing the conditional dependencies between the model inputs (white rectangle) and parameters (gray rounded rectangle) which need to be estimated. The white rounded rectangle are computed based on the estimations. The model can be interpreted as an empirical forward model (direction of arrows) where parameters affect the latent variables which in turn affect the data. Reconstructions are then obtained by inverting the model (i.e. computing the posterior) to obtain the latent variables given the data.

List of notations

| | |
|---|---|
| $\alpha$ | Concentrated parameter of the Dirichlet distribution (i.e. uncertainty) |
| $\beta$ | Regression coefficients |
| $\rho$ | Covariance matrix that determines the variation between and within fields |
| $\kappa$ | Scale parameter controls the range of spatial dependency |
| $\mu$ | Mean structures |
| $X$ | Spatial dependence residuals |
| $\eta$ | Latent Gaussian Markov random fields ($\eta = \mu + X$) |
| $Z_{\text{LCRs}}$ | Vector of proportions or realizations |
| $B_{\text{Covariates}}$ | Matrix of auxiliary datasets |
| $f$ | Link function (transforms from $\mathbb{R}^2$ to proportions $(0,1)^3$; $Z = f(\eta)$) |
| $Y_{\text{PbLCC}}$ | Observations, as proportions. |

**Table 1.** Summary of notation used in the Bayesian hierarchical model.

To account for observational uncertainty in the compositions, the PbLCC are modelled as draws from a Dirichlet distribution given concentrated parameter $\alpha$ (controlling the uncertainty) and the vector of proportions $\boldsymbol{Z}$,

$$\boldsymbol{Y}_{\text{PbLCC}}|\boldsymbol{Z}, \alpha \sim \text{Dir}(\alpha \boldsymbol{Z}) \qquad \alpha > 0, \boldsymbol{Z}_k \in (0,1), \sum_k \boldsymbol{Z}_k = 1.$$

To account for the spatial dependence in the proportions, $\boldsymbol{Z}$ is modelled as a transformation, $f$, of a latent GMRF, $\boldsymbol{\eta}$:

$$\boldsymbol{Z} = f(\boldsymbol{\eta}) \qquad f : \mathbb{R}^2 \to (0,1)^3$$

$$\boldsymbol{Z}_k = \begin{cases} \dfrac{\exp(\boldsymbol{\eta}_k)}{1 + \sum_{i=1}^{2} \exp(\boldsymbol{\eta}_i)} & \text{for } k = 1, 2 \\[4mm] \dfrac{1}{1 + \sum_{i=1}^{2} \exp(\boldsymbol{\eta}_i)} & \text{for } k = 3 \end{cases}.$$

The inverse of $f$ is called the additive log-ratio transformation (alr, Aitchison, 1986), i.e. $\boldsymbol{\eta}_k = \log(\boldsymbol{Z}_k/\boldsymbol{Z}_3), k = 1, 2$. The alr transformation is the multivariate extension of a logit transformation.

The latent field is modelled with a mean structure $\boldsymbol{\mu}$ and a spatially dependent residual $\boldsymbol{X}$,

$$\boldsymbol{\eta} = \begin{bmatrix} \boldsymbol{\eta}_1 \\ \boldsymbol{\eta}_2 \end{bmatrix} = \begin{bmatrix} \boldsymbol{X}_1 \\ \boldsymbol{X}_2 \end{bmatrix} + \begin{bmatrix} \boldsymbol{\mu}_1 \\ \boldsymbol{\mu}_2 \end{bmatrix} = \boldsymbol{X} + \boldsymbol{\mu}.$$

Here $\boldsymbol{X}$ is GMRF with a separable covariance structure;

$$\boldsymbol{X}|\kappa, \boldsymbol{\rho} \sim \text{N}(0, \boldsymbol{\rho}^{-1} \otimes \boldsymbol{Q}(\kappa))$$

where $\boldsymbol{Q}(\kappa)$ is the precision matrix of a spatially dependent GMRF (Lindgren et al., 2011), $\kappa$ is the scale parameter which controls the range of spatial dependency and $\boldsymbol{\rho}$ controls the variation within and between the fields $\boldsymbol{X}_k$ (see Pirzamanbein et al., 2015, for details).

The mean structure is modelled as a linear regression $\boldsymbol{\mu} = \boldsymbol{B}\boldsymbol{\beta}$, i.e. a combination of covariates $\boldsymbol{B}$ and regression coefficients $\boldsymbol{\beta}$. The main focus of this paper is to evaluate the model sensitivity to the choice of covariates (e.g. the auxiliary datasets). The PbLCC is modelled based on six different sets of covariates (Figure 1): 1) Intercept, 2) SRTM$_{\text{elev}}$, 3) K-L$_{\text{ESM}}$, 4) K-L$_{\text{RCA3}}$, 5) H-L$_{\text{ESM}}$, and 6) H-L$_{\text{RCA3}}$. Table 2 shows the different models and the corresponding covariates included in the model.

The model description is completed by specifying prior distributions for the model parameters. Wide but proper priors are assigned for $\alpha, \kappa, \boldsymbol{\rho}$ and $\boldsymbol{\beta}$. Specifically, a Gamma prior is chosen for the uncertainty and scale parameters, $\alpha$ and $\kappa$, i.e. $\Gamma(1.5, 0.1)$ and $\Gamma(1.5, 0.1)$. A Gaussian prior for the regression parameters $\boldsymbol{\beta}$, with zero expectation and small precision $q_\beta = 10^{-3}$. The $\boldsymbol{\rho}$ is assigned an inverse wishart prior, $IW(\mathbb{I}, 10)$, where $\mathbb{I}$ is a $2 \times 2$ identity matrix.

**2.2 Inference and associated uncertainties**

The Markov Chain Monte Carlo (MCMC) method (Brooks et al., 2011) is used to estimate the parameters and to reconstruct the land-cover composition, $\boldsymbol{Z}_{\text{LCRs}}$, with $100\,000$ MCMC samples and a burn-in sample size of $10\,000$. Details of the MCMC implementation can be found in Pirzamanbein et al. (2015).

| Model | Covariates | | | | | |
|---|---|---|---|---|---|---|
| | Intercept | SRTM$_{elev}$ | K-L$_{ESM}$ | K-L$_{RCA3}$ | H-L$_{ESM}$ | H-L$_{RCA3}$ |
| Constant | x | | | | | |
| Elevation | x | x | | | | |
| K-L$_{ESM}$ | x | x | x | | | |
| K-L$_{RCA3}$ | x | x | | x | | |
| H-L$_{ESM}$ | x | x | | | x | |
| H-L$_{RCA3}$ | x | x | | | | x |

**Table 2.** Six different models and corresponding covariates. SRTM$_{elev}$ is elevation (Becker et al., 2009), K/H indicates KK10 (Kaplan et al., 2009) or HYDE (Klein Goldewijk et al., 2011) land use scenarios and L$_{RCA3}$/L$_{ESM}$ indicates vegetation model driven by climate from the Rossby Centre Regional Climate Model (Samuelsson et al., 2011) or Earth System Model (Mikolajewicz et al., 2007).

In each MCMC iteration, the samples of $\boldsymbol{\eta}$ are obtained by adding the spatial dependency field $\boldsymbol{X}$ and the effect of covariates through $\boldsymbol{B\beta}$. Applying the alr transformation to the $\boldsymbol{\eta}$ samples, MCMC samples for $\boldsymbol{Z}$ are obtained. The land-cover reconstruction is then computed by averaging the MCMC samples, giving $\boldsymbol{Z}_{\text{LCRs}} = \text{E}(\boldsymbol{Z}|\boldsymbol{Y}_{\text{PbLCC}})$.

The uncertainties of the land-cover reconstruction, $\text{V}(\boldsymbol{Z}|\boldsymbol{Y}_{\text{PbLCC}})$, are assessed by constructing predictive regions (PR) using

5 the MCMC samples at each location. The predictive regions are constructed to represent the uncertainty associated with the reconstructions; including uncertainties in both model parameters ($\alpha$, $\kappa$, $\boldsymbol{\rho}$, and $\boldsymbol{\beta}$) and underlying fields ($\boldsymbol{Z}$). The predictive regions are constructed by first creating elliptical $95\%$-predictive regions (i.e. containing $95\%$ of the MCMC samples) in $\mathbb{R}^2$ (Figure 3 left plot) before transforming these to ternary predictive region in $(0,1)^3$ (Figure 3 right plot) (see Pirzamanbein et al., 2015, for details). In order to compare the uncertainties of different model land-cover reconstructions, we report the

10 fraction of the unit triangle covered by the ternary PR. This is done by distributing points in the ternary diagram and computing the fraction as the number of points laying inside the PR divided by total number of points in the ternary triangle.

**2.3 Testing the model performance**

To evaluate the model performance, we compared the land-cover reconstructions from different models for the 1900 CE time period with the European Forest Institute forest map (EFI-FM) by computing the average compositional distances (ACD). The

15 compositional distances between two different compositions, $\boldsymbol{U}$ and $\boldsymbol{V}$, are computed as (Aitchison et al., 2000)

$$\Delta(\boldsymbol{U} - \boldsymbol{V}) = \Delta(\boldsymbol{u} - \boldsymbol{v}) = \left( (\boldsymbol{u} - \boldsymbol{v})^\top \boldsymbol{H}^{-1} (\boldsymbol{u} - \boldsymbol{v}) \right)^{1/2}$$

where $\boldsymbol{u} = alr(\boldsymbol{U})$, $\boldsymbol{v} = alr(\boldsymbol{V})$ and $\boldsymbol{H}$ is a $2 \times 2$ matrix, neutralizing the choice of denominator in the alr transformation, with elements $H_{ij} = 2$ if $i = j$, and $H_{ij} = 1$ if $i \neq j$. These distances are then averaged over all locations. This measure is similar to root mean square error in $\mathbb{R}^2$ but it accounts for compositional properties, i.e. each component of the compositions is between

20 $(0,1)$ and sum of all the components is 1.

[Figure]

**Figure 3.** The left plot shows the $95\%$ elliptical predictive region in $\mathbb{R}^2$. The right ternary diagram shows the transformed $95\%$ predictive region together with the corresponding fraction, $60\%$, compared to the whole triangle.

Since no independent observational data exists for the 1725 CE and 4000 BCE time periods, we applied a 6-fold cross-validation scheme (Friedman et al., 2001, Ch. 7.10) for all six models and three time periods. The PbLCC data were divided into 6 randomly selected groups and, in each round, the distance between the left out data, $\boldsymbol{Y}_{\text{PbLCC},l}$, and predictions for group $l$ given a model fitted to the rest of the data, $\mathsf{E}(\boldsymbol{Z}_l|\boldsymbol{Y}_{\text{PbLCC},k}\ k\notin l)$, were computed.

5  To compare the predictive performance of the models, the Deviance Information Criteria (DIC; see Gelman et al., 2014, Ch. 7.2) is also computed for all models and time periods.

**3 Results and discussion**

Fossil pollen is a well-recognized information source of vegetation dynamics and generally accepted as the best observational data source on past land-cover composition and  environmental conditions (Trondman et al., 2015).

10  Today, central and northern Europe have, at the subcontinental spatial scale, the highest density of palynologically investigated sites on Earth. However, the collection of pollen data is very time consuming, cannot be performed everywhere and hardly applicable by climate modellers as input in their original format and therefore heavily underused. The lack of spatially explicit proxy based land cover data directly usable in climate models has been hampering the correct representation of past climate-land cover relationship.

15

Regrettably, the available auxiliary datasets (Table 2) exhibit large variation in the extent of coniferous and broadleaved forests, and un-forested areas for all of the studied time periods (Figure –1). These substantial differences illustrate large deviances between model based estimates of the past land-cover composition due to differences in climate forcing and/or applied ALCC scenarios. Differences in climate model outputs (e.g. Harrison et al., 2014; Gladstone et al., 2005) and ALCC model estimates (Gaillard et al., 2010) have been recognized in earlier comparison studies and syntheses. The effect of the differences in input climate forcing and ALCC scenario on DVM estimated land-cover composition presented here are especially pronounced for central and western Europe, and for elevated areas in northern Scandinavia and the Alps (Figure –1). In general the KK10 ALCC scenario produces larger un-forested areas, notably in western Europe, compared to the HYDE scenario. Compared to the ESM climate forcing; the RCA3 forcing results in higher proportions of coniferous forest, especially for central, northern and eastern Europe. The described differences are clearly recognizable for all the considered time periods and are generally larger between time periods than within each time period.

Usage of the above described, solely model based land cover to assess the impact of past anthropogenic changes on climate and terrestrial nutrient cycles has been shown to lead to largely diverging results (e.g. Strandberg et al., 2011). The importance of reliable land-cover representations for studying the biogeophysical impacts of anthropogenic land-cover change on climate is well recognized by the climate modelling community (e.g. Strandberg et al., 2011; Pitman et al., 2009; de Noblet-Ducoudré et al., 2012). Introducing the pollen based land cover reconstructions into climate models would facilitate the mechanistic studies on past climate-land cover relationships. The purpose of the statistical model described in Section 2.1 is to combine the observed PbLCC with the spatial structure in the auxiliary data to produce spatially complete maps of past land-cover that can be used directly in climate models.

To assess the impact of the different auxiliary datasets, the statistical model was used to create a set of  proxy based reconstructions of past land cover for central and northern Europe during three time periods (1900 CE, 1725 CE and 4000 BCE; see Figures 4–6). Each of the reconstructions were based on the irregularly distributed observed pollen data (PbLCC), available for ca $25\%$ of the area, together with one of the six models described in Table 2; each model uses a different combination of auxiliary data (Figure 1). The spatial dependence in the PbLCC data was modelled using GMRFs, along with additional spatial structure inferred from the auxiliary datasets (Pirzamanbein et al., 2015).

To illustrate the structure of the statistical model, step by step advancement from auxiliary data (model derived land cover) to final statistical estimates, for 1725 CE, are given in Figures 7 and 8. Figure 7 shows, for two locations, how the large differences in K-LRCA3 and K-LESM are reduced by scaling with the regression coefficients, $\beta$; capturing the empirical relationship between covariates and PbLCC data. The two land-cover estimates are then further subject to similar adjustments due to intercept and $SRTM_{elev}$, and finally similar spatial dependent effects. The corresponding decomposition of contributions to the continental map for 1725 CE, from $SRTM_{elev}$, K-$L_{ESM}$, and the spatial effects are given in Figure 8.

The final land-cover reconstructions achieved by fitting the models to the observed PbLCC are very similar, regardless of the auxiliary datasets used.

At first the similarity among the reconstructions might seem contradictory, but recall that the model allows for, and estimates, different weighting (the regression coefficients, $\beta$:s) for each of the auxiliary datasets. Thus, the resulting reconstruction do not

[Figure]

**Figure 4.** Land-cover reconstructions using pollen based land-cover compositions (PbLCC) for the 1900 CE time periods (top row). The reconstructions are based on six different models (see Table 2) with different auxiliary datasets. Locations and compositional values of the available PbLCC data are given by the black rectangles. Middle row shows the compositional distances between each model and the Constant model. Bottom row shows the compositional distances between each model and the EFI-FM.

[Figure]

**Figure 5.** Land-cover reconstructions using local estimates of pollen based land-cover compositions (PbLCC) for the 1725 CE time period (top row). The reconstructions are based on six different models (see Table 2) with different auxiliary datasets. Locations and compositional values of the available PbLCC data are given by the black rectangles. Bottom row shows the compositional distances between each model and the Constant model.

rely on the absolute values in the auxiliary datasets, only their spatial patterns; Table 3 illustrates the substantial discrepancies in the estimated coefficients, $\beta$.

| 1725 CE | Constant | Elevation | K-L$_{ESM}$ | K-L$_{RCA3}$ | H-L$_{ESM}$ | H-L$_{RCA3}$ |
|---|---|---|---|---|---|---|
| $\beta_{intercept,1}$ | -1.39 | -1.41 | -0.70 | -1.20 | -0.77 | -1.25 |
| $\beta_{intercept,2}$ | -1.01 | -1.01 | -0.73 | -1.05 | -0.69 | -1.01 |
| $\beta_{SRTM_{elev},1}$ | | 0.08 | 0.17 | 0.13 | 0.17 | 0.12 |
| $\beta_{SRTM_{elev},1}$ | | -0.16 | 0.07 | -0.01 | 0.08 | -0.02 |
| $\beta_{\bullet,1,1}$ | | | 0.01 | -0.07 | 0.02 | 0.02 |
| $\beta_{\bullet,1,2}$ | | | -0.02 | -0.01 | 0.01 | 0.13 |
| $\beta_{\bullet,2,1}$ | | | -0.01 | 0.10 | 0.03 | 0.15 |
| $\beta_{\bullet,2,2}$ | | | -0.19 | -0.17 | -0.18 | -0.12 |

**Table 3.** The estimated $\beta$ coefficient for the six models (see Table 2) for 1725 CE time period. The last four rows represents the coefficients for the modelled derived data set used in that model represented by $\beta_{\bullet,i,j}$:

[Figure]

**Figure 6.** Land-cover reconstructions using local estimates of pollen based land-cover compositions (PbLCC) for the 4000 BCE time period (top row). The reconstructions are based on six different models (see Table 2) with different auxiliary datasets. Locations and compositional values of the available PbLCC data are given by the black rectangles. Bottom row shows the compositional distances between each model and the Constant model.

[Figure]

**Figure 7.** Advancement of the model for two locations at 1725 CE. Starting from the value of the K-L$_{RCA3}$ and K-L$_{ESM}$ covariates (∗), the cumulative effects of regression coefficients, $\beta$, (+); the intercept and SRTM$_{elev}$ covariates (●); and, finally, the spatial dependency structures (○), are illustrated. With the final points (○) corresponding to the land-cover reconstructions, $Z_{LCRs}$, and ■ marking the observed pollen based land-cover composition.

[Figure]

**Figure 8.** Advancement of K-L$_{\mathrm{ESM}}$ models for the 1725 CE time period: (a) shows the effect of intercept and SRTM$_{\mathrm{elev}}$, (b) shows the mean structure, $\boldsymbol{\mu}$, including all the covariates, (c) shows the spatial dependency structure and finally (d) shows the resulting land-cover reconstructions, $\boldsymbol{Z}_{\mathrm{LCRs}}$, obtained by adding (b) and (c).

Although the land-cover reconstructions produced by different models are very similar, model performance for elevated areas and for the areas with low observational data coverage (e.g. eastern and south-eastern Europe) is improved by including covariates that exhibit distinct spatial structures for the given areas (Figures 4–6).

The resulting land-cover reconstructions exhibit considerably higher similarity with the PbLCC data than the auxiliary land-
5   cover datasets for all tested models and time periods (Figures 4–6). The predictive regions indicate the capability of all the models in capturing the PbLCC data and shows similar reconstruction uncertainties (Figure 9). Analogous to the reconstructions the uncertainty regions (the transformed ellipses are described in Figure 3) are very similar in both size and shape irrespective of the auxiliary dataset used. Further the PbLCC data almost always falls within the uncertainty region illustrating that the reconstructions are consistent with the data.

10    Although a temporal misalignment exists between the PbLCC data for the 1900 CE time period (based on pollen data from 1850 to the present) and the EFI-FM (inventory and satellite data from 1990-2005); EFI-FM provides the best complete and consistent land cover map of Europe for present time, making it a reasonable choice for the comparison. The main differences between the EFI-FM and the PbLCC data for the 1900 CE time period are: 1) lower abundance of broadleaved forests for most of Europe, 2) higher abundance of coniferous forest in Sweden and Finland, and 3) higher abundance of
15   unforested land in North Norway in the EFI-FM data than in the PbLCC data (Pirzamanbein et al., 2015). The average compositional distances computed between the land-cover reconstructions and the EFI-FM for 1900 CE show practically identical (1.47 to 1.48) distances between all six reconstructions and the EFI-FM, and small differences among the six presented models (Table 4). Note that the compositional differences for each grid cells are shown in Figures 4–6. Neither the DIC results (Table 5) nor the 6-fold cross validation results (Table 6) show any advantage among the six tested models for the different
20   time periods. Implying there is no clear preference among the models. These results clearly show that the developed statistical

[Figure]

**Figure 9.** The prediction regions and fraction of the ternary triangle covered by these regions are presented for three locations, the six models (see Table 2), and the 1900 CE, 1725 CE and 4000 BCE time periods. Construction and interpretation of the prediction regions are described in Section 2.2 and Figure 3.

interpolation model is robust to the choice of covariates. The model is suitable for reconstructing spatially continuous maps of past land cover from scattered and irregularly spaced pollen based proxy data.

ACD

| | | | 1900 CE | | | |
|---|---|---|---|---|---|---|
| Model | EFI-FM | Elevation | K-L$_{ESM}$ | K-L$_{RCA3}$ | H-L$_{ESM}$ | H-L$_{RCA3}$ |
| Constant | 1.47 | 0.06 | 0.19 | 0.17 | 0.19 | 0.18 |
| Elevation | 1.48 | | 0.18 | 0.16 | 0.18 | 0.17 |
| K-L$_{ESM}$ | 1.47 | | | 0.08 | 0.10 | 0.07 |
| K-L$_{RCA3}$ | 1.47 | | | | 0.06 | 0.11 |
| H-L$_{ESM}$ | 1.47 | | | | | 0.08 |
| H-L$_{RCA3}$ | 1.47 | | | | | |
| | | | 1725 CE | | | |
| Constant | | 0.11 | 0.19 | 0.16 | 0.19 | 0.18 |
| Elevation | | | 0.12 | 0.14 | 0.14 | 0.16 |
| K-L$_{ESM}$ | | | | 0.15 | 0.16 | 0.07 |
| K-L$_{RCA3}$ | | | | | 0.08 | 0.18 |
| H-L$_{ESM}$ | | | | | | 0.17 |
| | | | 4000 BCE | | | |
| Constant | | 0.12 | 0.19 | 0.21 | 0.21 | 0.23 |
| Elevation | | | 0.12 | 0.19 | 0.16 | 0.21 |
| K-L$_{ESM}$ | | | | 0.19 | 0.21 | 0.07 |
| K-L$_{RCA3}$ | | | | | 0.07 | 0.19 |
| H-L$_{ESM}$ | | | | | | 0.21 |

**Table 4.** The average compositional distances among the six models (see Table 2) fitted to the data for each of the three time periods.

| DIC | 1900 CE | 1725 CE | 4000 BCE |
|---|---|---|---|
| Constant | **-562** | -656 | -591 |
| Elevation | -557 | -668 | -590 |
| K-L$_{ESM}$ | -551 | -654 | -601 |
| K-L$_{RCA3}$ | -559 | **-673** | -588 |
| H-L$_{ESM}$ | -554 | -654 | **-607** |
| H-L$_{RCA3}$ | -559 | -672 | -594 |

**Table 5.** Deviance information criteria (DIC) for each of the six models (see Table 2) and three time periods.

| ACD | 1900 CE | 1725 CE | 4000 BCE |
|---|---|---|---|
| Constant | 0.98 | 1.13 | 1.19 |
| Elevation | 0.98 | 1.11 | 1.20 |
| K-L$_{ESM}$ | 0.99 | 1.12 | 1.18 |
| K-L$_{RCA3}$ | 0.99 | 1.13 | 1.18 |
| H-L$_{ESM}$ | 1.00 | 1.12 | **1.17** |
| H-L$_{RCA3}$ | **0.97** | **0.97** | 1.17 |

**Table 6.** Average compositional distances from 6-fold cross-validations for each of the six models (see Table 2), and three time periods.

**4 Conclusions**

The  statistical model and Bayesian interpolation method presented here has been specially designed for handling palaeo-proxy records and, dependent on proxy data availability, is globally applicable. The ability of the model to create pollen based land cover reconstructions at sub-continental scale was illustrated using an example application on two Holocene time periods frequently assessed by climate researchers: the Little Ice Age (1725 CE) and the Holocene Thermal Maximum (4000 BCE); as well as a third time period covering the recent past (1900 CE) which was used for model validation. The model combines irregularly distributed, pollen based estimates of land cover representing 25% of the study area. , with auxiliary data and estimates spatial dependencies to produce land-cover maps. The resulting maps capture important features in the pollen proxy data and are reasonably insensitive to the use of different auxiliary datasets.

The considered auxiliary datasets were complied using most commonly utilized sources of the past land-cover data (estimates produced by a dynamic vegetation model and anthropogenic land-cover changes scenarios). These datasets exhibit considerable model and/or input dependant differences in their recreation of the past land cover. Emphasizing the need for the independent,  proxy based past land-cover maps created in this paper.

The  results also indicate that the model has a very good performance and will be very useful for large-scale, continental reconstructions of past land cover. The spatial model tested in this paper can provide an important tool to generate the regional to global scale land-cover maps based on proxy data. Such, pollen based past land cover reconstructions with global coverage are currently produced by the PAGES (Past Global changES) LandCover6k initiative [1] for most of globe.

The model's sensitivity to usage of different auxiliary datasets was validated by calculating deviance information criteria (DIC) and using cross validation for all the time periods. For the recent time period, 1900 CE, the land-cover reconstructions from the different models were also compared against a present day forest map. The evaluation indicates that the applied statistical model is robust. The model estimates the empirical relationship between auxiliary data and pollen based proxy data, therefore, it only considers features in the auxiliary data that are consistent with the  proxy data. The covariates with detailed spatial information improves the interpolation results for areas with low  proxy data coverage, however the overall performance remains unchanged.

Therefore, the model can provide reliable results using a variety of land cover data sets that capture important spatial patterns from vegetation models and past human land use, absent good covariates elevation can be used as the only auxiliary dataset. An important feature of the suggested model is the estimation of different weights for each of the auxiliary datasets (see Table 3), thus capturing the spatial patterns and not the absolute values in the auxiliary datasets. Our validations indicate that auxiliary datasets obtained using different climatic drivers produce very similar reconstructions, which are all close to the pollen based proxy data.

This modelling approach has strong ability to produce empirically based land-cover reconstructions for climate modelling purposes. Such reconstructions are necessary to evaluate the anthropogenic land-cover change scenarios currently used in the climate modelling and to study interactions between land cover and climate in the past with greater reliability.

**5 Data availability**

The database containing the reconstructions of coniferous forest, broadleaved forest and un-forested land, three fractions of land cover, for the three time-periods presented in this paper, along with reconstructions for 1425 CE and 1000 BCE using only the K-L$_{ESM}$ are available for download from https://behnaz.pirzamanbin.name/phd/.

*Acknowledgements.* The research presented in this paper is a contribution to the two Swedish strategic research areas Biodiversity and Ecosystems in a Changing Climate (BECC), and ModElling the Regional and Global Earth system (MERGE).

Lindström has been funded by Swedish Research Council (Vetenskapsrådet) grant no 2012-5983.

The authors would like to acknowledge Marie-José Gaillard for her efforts in providing the pollen based land-cover  proxy data and thank her for valuable comments on this manuscript.
* * *
[1]www.pastglobalchanges.org/ini/wg/landcover6k/intro